# Ellipse Detection with Applications of Convolutional Neural Network in Industrial Images

Kang Liu [1] , Yonggang Lu [2], Rubing Bai [2], Kun Xu [2], Tao Peng [1], Yichun Tai [1] and Zhijiang Zhang [1,*]

1   School of Communication and Information Engineering, Shanghai University, Shanghai 201900, China; liuk@shu.edu.cn (K.L.); 17820125peng@shu.edu.cn (T.P.); taiyc@shu.edu.cn (Y.T.)
2   Metallurgical Baosteel Technical Services Co., Ltd., Shanghai 201900, China; luyong@163.com (Y.L.); bairub@163.com (R.B.); kunxu2008@hotmail.com (K.X.)
*   Correspondence: zjzhang@staff.shu.edu.cn

**Abstract:** Ellipse detection has a very wide range of applications in the field of industrial production, especially in the geometric detection of metallurgical hinge pins. However, the factors in industrial images, such as small object size and incomplete ellipse in the image boundary, bring challenges to ellipse detection, which cannot be solved by existing methods. This paper proposes a method for ellipse detection in industrial images, which utilizes the extended proposal operation to prevent the loss of ellipse rotation angle features during ellipse regression. Moreover, the Gaussian angle distance conforming to the ellipse axioms is adopted and combined with smooth $L_1$ loss as the ellipse regression loss function to enhance the prediction accuracy of the ellipse rotation angle. The effectiveness of the proposed method is demonstrated on the hinge pins dataset, with experiment results showing an $AP_*$ of 80.93% and indicating superior detection performance compared to other methods. It is thus suitable for engineering applications and can provide visual guidance for the precise measurement of ellipse-like mechanical parts.

**Keywords:** ellipse detection; convolutional neural network; hinge pins; proposal extension; Gaussian angle distance

## 1. Introduction

In the metallurgical industry, heavy-duty conveyors are used, and the core component of these conveyors is the chain, which is made up of several hinge pins connected with multiple link plates, as shown in Figure 1. During long-term material transportation, the connections between the hinge pins and link plates are subject to severe wear and corrosion due to factors such as friction and humidity. Over time, adjacent hinge pins gradually deviate from their initial positions. When the deviation reaches the twisting limit of the hinge pins, the entire chain breaks at that point, thus affecting production [1]. Therefore, it is necessary to adopt an automated visual inspection method for the measurement of the spacing between adjacent hinge pins in the chain.

Given that the hinge pins are projected as elliptical shapes in the image, we need to perform ellipse object detection on the hinge pins. Ellipse detection methods can be broadly categorized into traditional methods and deep-learning-based methods. Traditional ellipse detection methods, such as Hough-transform-based methods, have high computational costs and are very time-consuming [2]. The least-squares-based methods [3] extract ellipses by fitting edge pixels to a general conic. However, this approach cannot disregard potential outliers within a set of edge pixels, making it susceptible to noise. Utilizing the connectivity between edge pixels, the edge-following methods detect ellipses [4], but their operation at the arc level leads to relatively lower reliability in detecting incomplete ellipses. Therefore, these methods are not suitable for ellipse detection in industrial images.

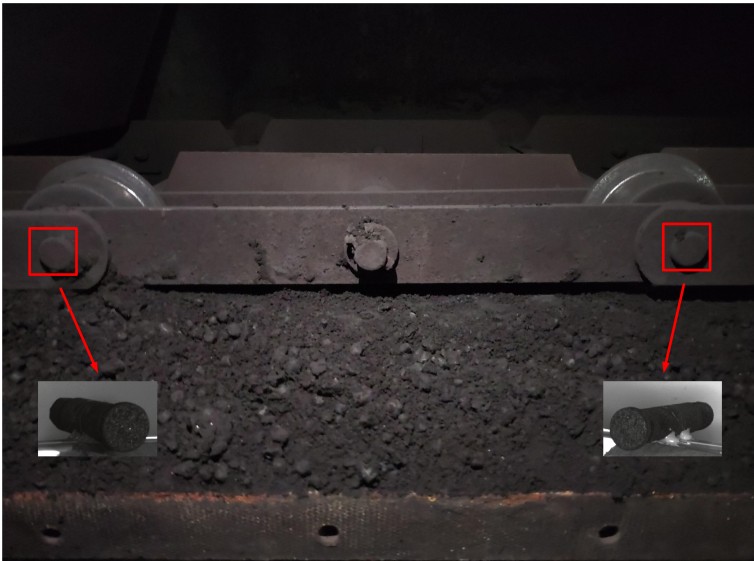

**Figure 1.** The chain is composed of hinge pins and link plates in the metallurgical sites.

With the rapid development of deep learning, the application of object detection models based on convolutional neural networks (CNNs) in ellipse detection has become a popular research direction [5–7]. Compared to traditional methods, these methods have the advantages of higher accuracy and greater robustness to environmental noise. Therefore, CNN-based methods can be employed for ellipse detection in industrial images. Considering the requirements for detection accuracy and stability in industrial environments, we use the two-stage classic object detection model Mask R-CNN as the baseline model for the proposed network.

In this paper, we present a robust and simplified ellipse regression model that is capable of detecting and parameterizing individual ellipse objects. We discard the mask prediction branch of the Mask R-CNN model and replace bounding box regression with ellipse parameter regression. During regression, considering the efficiency of the detection model, we only employ an extension proposal operation from Ellipse R-CNN to prevent the loss of angle information. In contrast, in selecting a suitable loss function for our ellipse regression model, we opt for the Gaussian angle distance, which adheres closely to the metric axioms of the ellipse. However, depending exclusively on the Gaussian angle distance as the loss function may lead to inaccurate estimations of local parameters, such as the ellipse rotation angle in certain situations. To address this issue, we combine the smooth $L_1$ loss function to further reduce the error in the regression of the ellipse angle. The contributions of this paper are summarized as follows:

- We propose a CNN-based method for ellipse object detection and apply it specifically to the detection of special component objects, such as the hinge pins in the metallurgical industry. By utilizing the proposed models, we can accurately detect the elliptical shape of the hinge pins in the images.
- We employ the extended proposal operation to address the issue of losing the rotation angle direction of the ellipse. Additionally, the Gaussian angle distance function and smooth $L_1$ loss function are combined as the loss function for the ellipse parameter regression task.
- We create a labeled small-scale dataset of hinge pins and conduct experiments related to this research by using the dataset. We validated its accuracy and robustness by comparing our method with traditional methods and other CNN-based approaches.

The remainder of this paper is organized as follows. Section 2 reviews the state-of-the-art methods in related work. Our method is described in detail in Section 3. Section 4 provides experimental validation of the superiority of our method from various perspectives. Section 5 concludes the paper.

## 2. Review of Related Work

In this section, we review some existing CNN-based object detection methods. In addition, we also review some metric methods that conform to the ellipse axioms for use as loss functions.

### 2.1. CNN-Based Object Detection Methods

Object detection has been a challenging task in the field of computer vision for a long time, aiming to automatically locate and recognize objects in the image. In recent years, using CNN-based methods for object detection has gradually become mainstream. These methods are mainly divided into one-stage detection and two-stage detection. One-stage detection methods are represented by networks such as RetinaNet [8], SSD series [9,10], and YOLO series [11,12]. Different from such methods, the two-stage methods require an additional step to generate proposals.

Many researchers have proposed various detection models for the two-stage methods, such as the Faster R-CNN [13]. The Faster R-CNN introduces the region proposal network (RPN) module, which learns to propose object regions. However, it encounters some difficulties in detecting small objects and highly occluded objects. Building upon the improvement of Faster R-CNN, He et al. [14] propose Mask R-CNN, which integrates a branch for object mask prediction. The region of interest (RoI) Pooling layer in Faster R-CNN is also replaced with RoIAlign, which leads to an even greater improvement in detection accuracy. Based on Mask R-CNN, Cheng et al. [15] propose BMask R-CNN, which introduces a boundary preservation mechanism, achieving more accurate capturing of object instance boundary information. However, the addition of the binary mask branch increases the computational cost of the network. In addition, Dai et al. [16] introduce the R-FCN network. It transforms the object detection problem into a pixel-level classification task. R-FCN utilizes the fully convolutional network (FCN) to densely perform pixel-level classification, enabling it to better leverage spatial information and enhance the accuracy of object localization.

For specific objects, such as ellipses, there are also researchers conducting relevant studies. Dong et al. [17] introduce an improved Ellipse R-CNN network based on Mask R-CNN. This method uses a novel proposal extension method that can better address the issue of uncertainty in ellipse rotation. However, its accuracy may be affected by the variation in the shape and pose of the ellipse. Loncomilla et al. [18] propose Rocky-CenterNet for rock detection, using the ellipse to enclose the boundary of rocks to better describe their shapes. This approach demonstrates higher adaptability and precision in handling irregularly shaped objects compared to traditional bounding boxes. However, as this method employs the ellipse as the bounding box for objects, it may result in the loss of boundary information. Oh et al. [19] employ a CNN to detect elliptical LED markers. They utilize the predicted ellipse rotation angle as a measure of uncertainty in CNN predictions, achieving robust detection of LED markers without the need for adjusting feature extraction parameters. However, the detection and recognition accuracy of these markers can be affected when they are obstructed. Dong et al. [20] propose an ellipse detection network based on domain randomization techniques. They build a detector with rotation filters and a rotation region proposal network to accurately detect ellipses. However, since the training data is generated through a virtual environment, its generalization performance in real-world scenarios requires further validation.

### 2.2. Loss Functions

In the task of bounding box regression, there are many metrics such as the loss functions for measuring the distance between the ground-truth bounding box and the proposal. In R-CNN [21] and SPPNet [22], the smooth $L_2$ loss is used as the loss function for bounding box regression. In Fast R-CNN [23], the smooth $L_1$ loss is adopted as it is less sensitive to outliers. For the ellipse regression task, relying solely on smooth $L_1$ or $L_2$

loss is insufficient to effectively complete the regression of ellipse parameters. Thus, it is necessary to explore other distance metric methods.

Zhou et al. [24] propose a method for representing the ellipse parameters of objects in arbitrary orientations. It employs a two-dimensional Gaussian distribution label assignment for coarse sample selection, followed by the use of Kullback–Leibler divergence (KLD) loss to refine the coarse samples. However, it should be noted that KLD is asymmetrical, meaning that the distance between two ellipse Gaussian distributions cannot be computed interchangeably. Li et al. [25] propose a shape-biased ellipse detection network with an auxiliary task. In terms of the loss function, the introduction of the Wasserstein distance further enhances the precision of ellipse detection. However, the Wasserstein distance has a high computational complexity, limiting the efficiency of the network in practical applications. Llerena et al. [26] propose modeling object bounding boxes as two-dimensional Gaussian distributions and introduce the Hellinger distance for similarity measurement of ellipse representations, which improves the accuracy of object detection. However, due to the influence of the Hellinger distance, the model is sensitive to noise in regions where the distribution has small values.

In lunar crater identification [27], Christian et al. present a novel distance metric method referred to as the Gaussian angle distance. This distance metric is built upon an ellipse matrix, which is interpreted as a binary Gaussian function. It satisfies ellipse axioms such as symmetry and can be directly analyzed and calculated by utilizing the respective parameters of the two ellipses being compared. However, the Gaussian angle distance only considers the angle relationship between two distributions and does not take into account the magnitude, which will limit its applicability in certain tasks.

## 3. Proposed Method

### 3.1. Ellipse Regression

The conventional procedure of object detection methods utilizing Mask R-CNN encompasses several sequential steps. Initially, the backbone, exemplified by the ResNet-50 network, extracts image features. Subsequently, the proposals generated by the RPN are partitioned into distinct scales, which are then passed to the feature pyramid network (FPN) [28] to generate feature maps at varying scales. These feature maps are uniformly cropped using the RoIAlign layer, resulting in feature maps of equal dimensions for tasks such as classification, box regression, and object mask prediction.

In our ellipse detection task, we only focus on ellipse parameter regression and classification, so the mask prediction branch can be discarded. The overall framework of our network is shown in Figure 2. From Figure 2, the traditional bounding box regression is replaced with ellipse parameter regression. The results of box regression are the center coordinates, width, and height of the bounding box. In contrast, ellipse regression requires the regression of five parameters: the center coordinates $(x_0, y_0)$, the semi-major and semi-minor axes $a, b (a \geq b)$, and the rotation angle $\theta$ (measured from the positive $x$-axis to the semi-major axis of the ellipse). These five parameters uniquely define an ellipse. The equation of a general ellipse can be expressed using these parameters as follows:

$$\frac{(x' \cos \theta + y' \sin \theta)^2}{a^2} + \frac{(-x' \sin \theta + y' \cos \theta)^2}{b^2} = 1, x' = x - x_0, y' = y - y_0, \tag{1}$$

where the ellipse orientation is $\theta \in (-\frac{\pi}{2}, \frac{\pi}{2}]$. In the ellipse regression, there are the ground-truth of the ellipse parameters $E = (E_x, E_y, E_a, E_b, E_\theta)$, which separately denote the center coordinates, semi-major axis, semi-minor axis and rotation angle of the ellipse ground-truth. The ellipse proposal is $P = (P_x, P_y, P_w, P_h)$, where the first two parameters represent the ellipse center coordinates and the last two parameters denote the width and height of the ellipse proposal.

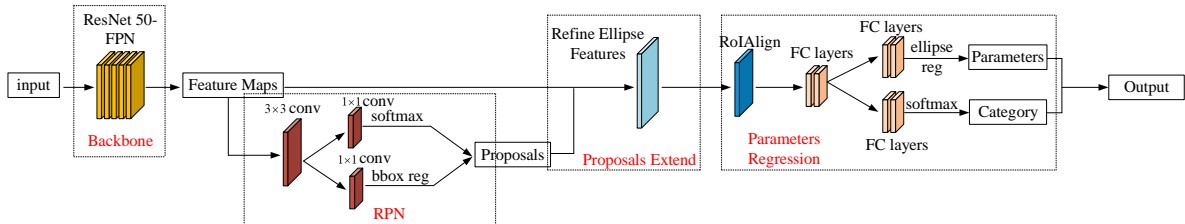

**Figure 2.** The overall framework of our network.

Compared to bounding box regression, ellipse regression differs not only in the number of regression parameters but also in that its directional information is more prone to loss during the regression process of the incomplete ellipse at the image boundary, as illustrated in Figure 3. Once the RPN generates proposals of different sizes, they are sent to the RoIAlign layer and adjusted to a fixed size, causing the feature map to become distorted and rendering the prediction of the original ellipse's orientation information unstable. From Ellipse R-CNN [17], we can learn that when performing ellipse parameter regression, the ellipse proposal $P$ can be extended into a square area $Q$. The extension area is $Q = (Q_x, Q_y, Q_l)$, where $(Q_x, Q_y) = (P_x, P_y)$ is the center coordinates of the extended proposal, and $Q_l = \sqrt{P_w^2 + P_h^2}$ is the square length of the extended proposal.

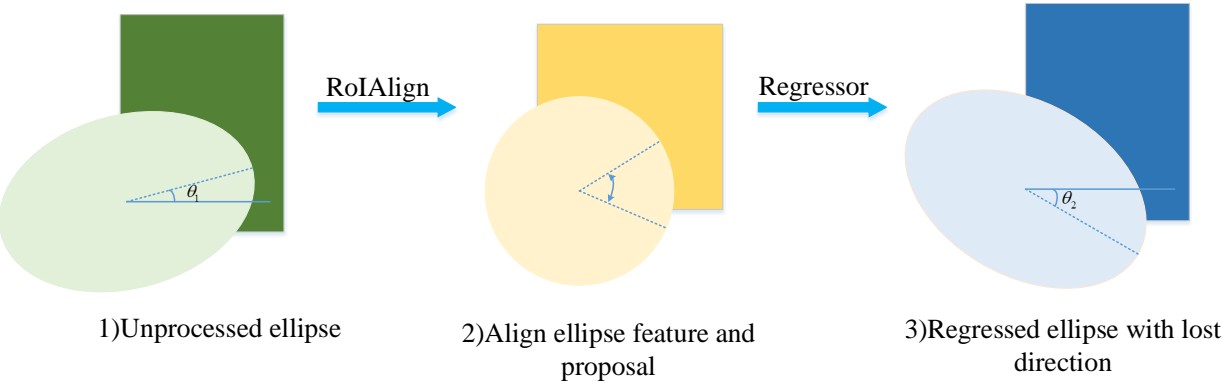

1)Unprocessed ellipse  2)Align ellipse feature and proposal  3)Regressed ellipse with lost direction

**Figure 3.** The proposals generated by RPN will be adjusted to a fixed-size square after passing through the RoIAlign layer, and then ellipse parameter regression is performed with the arbitrary ellipse rotation angle feature. It will result in an unstable prediction of the orientation of the regressed ellipse.

As shown in Figure 4, when the detected ellipse is located at the image boundary, it is possible to occur an incomplete ellipse on the image. During the ellipse parameter regression process, if only the five parameters of the ellipse are regressed, the shape of the ellipse may not be accurately regressed due to the influence of the incomplete proposals generated by the RPN module. To prevent this situation from occurring, an additional visibility parameter $s = \frac{Q_l}{E_l}$ needs to be regressed to indicate the visibility ratio of the incomplete ellipse on the image, where $s \in (0, 1]$, and $E_l = 2\sqrt{E_a^2 + E_b^2}$ is the square length of enclosing the ellipse $E$. The higher value of $s$ indicates the closer match between the extended proposal $Q$ and the ground truth $E$, as well as a higher visibility ratio of the detected ellipse. When the value of $s$ is 1, it indicates that the detected ellipse appears completely in the image. Based on the value of parameter $s$, this regression process can adapt to the detection of all ellipses.

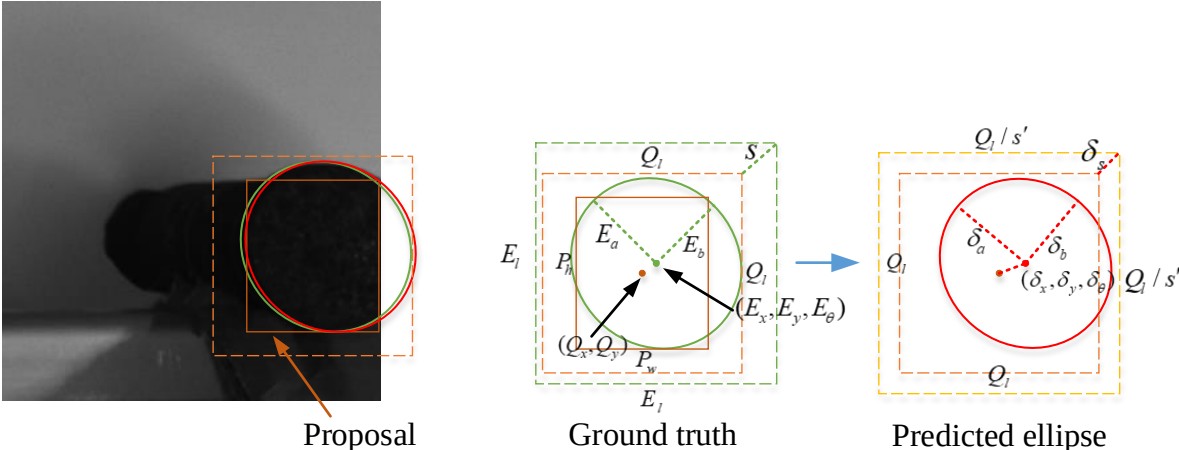

**Figure 4.** The process of ellipse parameters regression.

We can associate the other five predicted offset parameters of the ellipse with this scaling factor. Therefore, we can regress six relative offset parameters $\delta_x, \delta_y, \delta_a, \delta_b, \delta_\theta, \delta_s$, and the specific expression is as follows:

$$\begin{cases} \delta_x = s'(E'_x - Q_x)/Q_l, \\ \delta_y = s'(E'_y - Q_y)/Q_l, \\ \delta_a = log(2s'E'_a/Q_l), \\ \delta_b = log(2s'E'_b/Q_l), \\ \delta_\theta = E'_\theta/\pi, \\ \delta_s = log((s'+1)/2), \end{cases} \tag{2}$$

$$\begin{cases} \delta^*_x = s(E_x - Q_x)/Q_l, \\ \delta^*_y = s(E_y - Q_y)/Q_l, \\ \delta^*_a = log(2sE_a/Q_l), \\ \delta^*_b = log(2sE_b/Q_l), \\ \delta^*_\theta = E_\theta/\pi, \\ \delta^*_s = log((s+1)/2), \end{cases} \tag{3}$$

where $\delta^*$ is the ellipse regression relative offset parameters ground truth, $E'$ is the predicted ellipse parameters, and $s'$ is the predicted visibility ratio.

After obtaining the relative offset parameters, the ellipse parameters can be predicted, as shown below:

$$\begin{aligned} & E'_x = \frac{Q_l}{s'}\delta_x + Q_x, E'_y = \frac{Q_l}{s'}\delta_y + Q_y, s' = 2exp(\delta_s) - 1, \\ & E'_a = \frac{Q_l}{2s'}exp(\delta_a), E'_b = \frac{Q_l}{2s'}exp(\delta_b), \theta' = \pi\delta_\theta, \\ & E'_\theta = \begin{cases} atan2(sin\,\theta', cos\,\theta'), if\,cos\,\theta' \geq 0 \\ atan2(-sin\,\theta', -cos\,\theta'), if\,cos\,\theta' < 0, \end{cases} \end{aligned} \tag{4}$$

where the rotation angle $E'_\theta \in (-\frac{\pi}{2}, \frac{\pi}{2}]$.

### 3.2. Improved Loss Function

In Faster R-CNN, the smooth $L_1$ loss function is used to predict the parameters offsets between the bounding box and the ground-truth. However, in the task of ellipse detection, encompassing six parameters, these loss functions are no longer appropriate for detection.

We can represent an ellipse as a matrix, denoted as $A_i$. Suppose there are two ellipses in an image, $A_i$ and $A_j$. When the two ellipses are not identical, there will be a relative distance between them. Due to the uniqueness of ellipses as geometric shapes, specific axioms are required to describe their relative distance relationships [27,29], as follows:

1.  Minimality: $d(A_i, A_j) = 0$ when $A_i = A_j$. Indicates that the distance between two ellipses is zero.
2.  Symmetry: $d(A_i, A_j) = d(A_j, A_i)$. When $A_i$ and $A_j$ are swapped with each other, the distance between them does not change.
3.  Triangle Inequality: $d(A_i, A_j) \leq d(A_i, A_k) + d(A_k, A_j)$. When there is a third ellipse $A_k$, the distances between them satisfy the triangle inequality.
4.  Similarity Invariance: $d(A_i, A_j) = d(S[A_i], S[A_j])$, where $S[\cdot]$ is a similarity transformation. This indicates that the two ellipses undergo the same similarity transformation such as rotation, translation, and scaling in the image, and their distance should remain the same.

The ellipse matrix can be interpreted as a binary Gaussian probability distribution. Hence, various distance metrics can be used between two probability distributions, such as the KLD, the Wasserstein distance, and the Gaussian angle distance. However, not all methods satisfy the above axioms. For instance, the KLD does not satisfy the triangle inequality and is also highly unstable when the distance between two distributions is small or large [30], making it unsuitable for ellipse parameter regression. Furthermore, while the Wasserstein distance satisfies the first three required axioms, it does not satisfy the Similarity Invariance axiom.

Therefore, we can choose the Gaussian angle distance as the loss function, which can satisfy the above four axioms. The Gaussian angle distance between a ground-truth ellipse matrix $A_E$ and a predicted ellipse matrix $A_{E'}$ is given by [27]:

$$d_{GA}(A_E, A_{E'}) = arccos\left\{ \frac{4\sqrt{|Y_E||Y_{E'}|}}{|Y_E + Y_{E'}|} exp[-\tfrac{1}{2}(y_E - y_{E'})^T Y_E (Y_E + Y_{E'})^{-1} Y_{E'}(y_E - y_{E'})] \right\}, \tag{5}$$

where $2 \times 2$ submatrix $Y_E$ is as follows:

$$Y_E = \begin{bmatrix} cosE_\theta & -sinE_\theta \\ sinE_\theta & cosE_\theta \end{bmatrix} \begin{bmatrix} 1/E_a^2 & 0 \\ 0 & 1/E_b^2 \end{bmatrix} \begin{bmatrix} cosE_\theta & sinE_\theta \\ -sinE_\theta & cosE_\theta \end{bmatrix}, \tag{6}$$

and $y_E^T = \begin{bmatrix} E_x & E_y & 1 \end{bmatrix}$ is the center homogeneous coordinate of the ellipse. It is the same as the expression of $Y_{E'}$ and $y_{E'}^T$. It can be seen from the above formula that the method can be analyzed and calculated according to the parameters of the two ellipses $A_E$ and $A_{E'}$ in the image.

For typical ellipse regression tasks, the Gaussian angle distance can be an effective choice as the loss function. However, its suitability might vary in specific scenarios. As shown in Figure 5, when the major and minor axes of the ellipse are close, it means that the ellipse can be approximated as a standard circle. In such cases, employing the Gaussian angle distance as the distance metric for ellipse regression can yield similar distance values for distinct orientations of predicted ellipses and the ground-truth ellipse. This behavior is attributed to the property of the Similarity Invariance exhibited by the Gaussian angle distance. Irrespective of the ellipse's orientation, their Gaussian angle distance tends to be proximate if the cross-overlap area ratio remains alike. Although the overall performance of the two predicted ellipses could exhibit similarity, subtle distinctions within the internal ellipse parameters, notably the ellipse rotation angle, can pose challenges for accurate orientation regression.

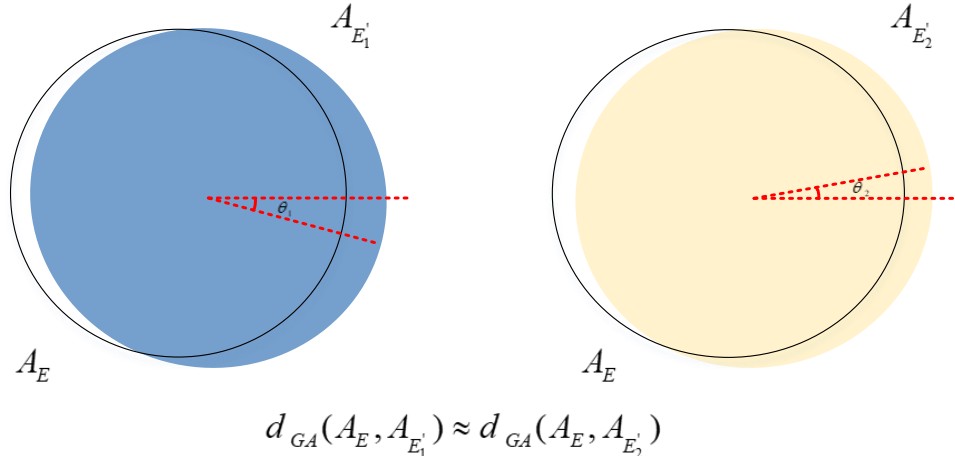

$$d_{GA}(A_E, A_{E_1'}) \approx d_{GA}(A_E, A_{E_2'})$$

**Figure 5.** When the ground-truth ellipse is approximated as a standard circle, the predicted ellipses with different orientations can have similar Gaussian angle distance values.

In Faster R-CNN, Ren employs the smooth $L_1$ loss function for object detection regression, predicting the four bounding box parameters. Similarly, we can utilize the smooth $L_1$ loss function for ellipse prediction. However, as previously discussed, relying solely on the smooth $L_1$ loss function for ellipse regression yields suboptimal results. Hence, our approach involves employing the Gaussian angle distance as the primary loss function for comprehensive ellipse parameter regression. Furthermore, in light of the aforementioned challenge regarding accurate rotation angle regression in specific scenarios, we introduce the smooth $L_1$ loss function as a supplementary element to enhance rotation angle prediction. Based on the above theoretical description, we can design the following ellipse regression loss function expression:

$$\mathcal{L}_e = d_{GA}(A_E, A_{E'}) + \alpha \mathcal{R}\left(\frac{E_\theta}{\pi} - \frac{E_\theta'}{\pi}\right), \tag{7}$$

where $\mathcal{R}$ is the smooth $L_1$ loss function and weight factor $\alpha$ represents the ratio of the smooth $L_1$ loss between the ground-truth angle and the predicted angle in the loss function. In the subsequent experimental process, we set $\alpha = 2$.

## 4. Experimental Results

In Section 1, we briefly introduce the necessity of detecting hinge pins in metallurgical sites. In this section, we conduct some experiments based on the ellipse hinge pins detection task using the network model we propose and verify its superiority compared with other models.

### 4.1. Hinge Pins Dataset

Due to the limitations in the industrial environment, we use the hinge pins fixed to a movable guide rail to simulate the chain-driven state in a real scene for image acquisition. The scene is shown in Figure 6. For hardware selection in image acquisition, we use the MER-502-79U3M model camera with dimensions of 2048 × 2048. The camera lens used is the LM5JC10M lens with a focal length of 5 mm. A total of 1862 images of the hinge pins are collected, taken from different angles and distances. Considering the requirement of our network's image input being 512 × 512, we divide the original images with dimensions of 2048 × 2048 into smaller patches by dividing them into equal quarters in width and height. One original image can yield 16 smaller images with dimensions of 512 × 512, among which one to two images contain the hinge pins object we want to detect. After processing all images, a dataset of hinge pins, containing 3317 new images, can be obtained.

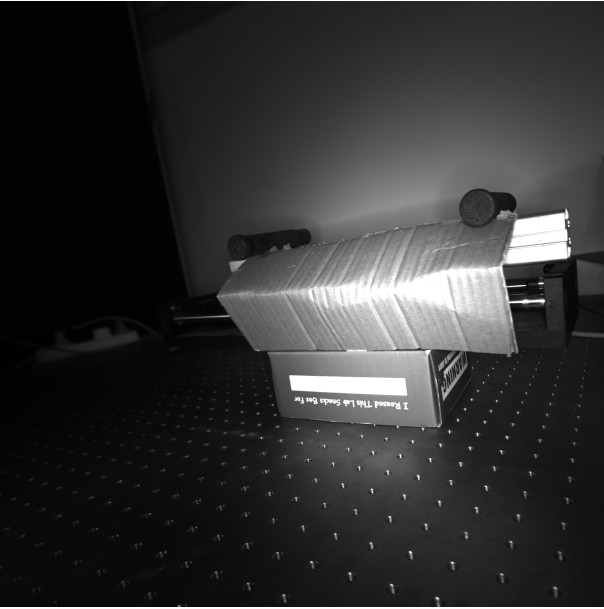

**Figure 6.** The hinge pins are fixed on the guide rail.

Then, these images are subjected to manual annotation. The edges of the hinge pins are annotated using the Labelme annotation tool, based on the theoretical foundation of fitting ellipses to edge point sets. For each ellipse, five edge points are marked and the ellipse parameters $(E_x, E_y, E_a, E_b, E_\theta)$ are obtained through ellipse fitting, where $E_x$ and $E_y$ are the coordinates of the center of the ellipse, $E_a$ and $E_b$ are the semi-major axis and semi-minor axis of the ellipse, and $E_\theta$ is the rotation angle of the ellipse. The obtained ellipse parameters are stored in JSON annotation files, corresponding with the ground-truth images. The dataset is divided into training and testing sets following a ratio of 0.9:0.1. Some example images with the annotated ellipse are shown in Figure 7.

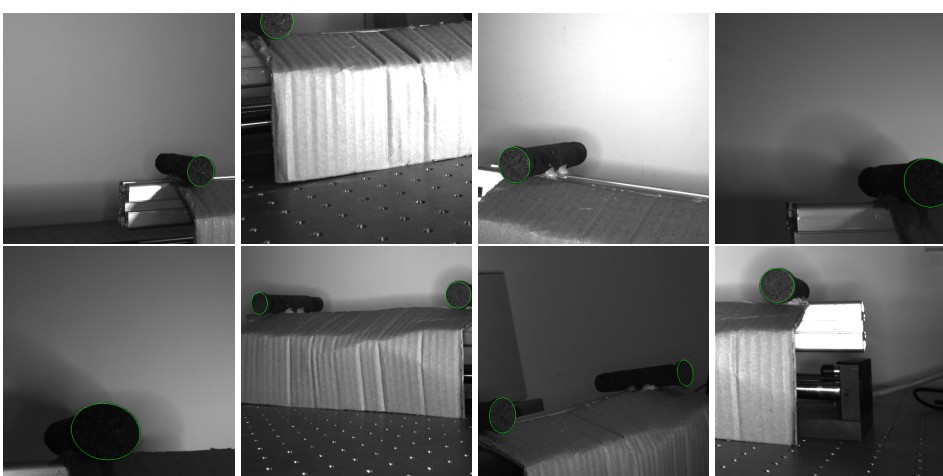

**Figure 7.** The examples of hinge pins with the annotated ellipse.

In the actual detection process, to accurately determine the specific position of the hinge pins in the original image, the original images are divided into 16 equal patches, and each patch is assigned a unique ID from 0 to 15 according to the partition sequence. During training, patches without objects can be excluded from the training process. Each image is also divided into 16 patches with corresponding IDs during inference. By predicting confidence scores and setting a threshold, patches with scores above the threshold are selected, and their IDs are used to map the predicted ellipse parameters back to the original image at their fixed positions. This process allows us to obtain the actual ellipse parameters in the original image.

*4.2. Experimental Setup*

The experiments are executed on a server equipped with the Ubuntu 16.04 operating system. The server has a Xeon Silver 4216 CPU, four GTX 2080Ti GPUs, and 256-GB memory. We train 60 epochs on the training sets, and the model with the lowest verification loss is saved for testing, with batch size 16, momentum 0.9, learning rate 0.005, and weight decay 0.0001. Based on the Mask R-CNN network structure, we use the resnet-50 pre-trained model to extract ellipse features and employ PyTorch lightning to train our model. We apply some evaluation metrics in our experiments to evaluate the detection performance of our model versus other models, including mean intersection over union (MeanIOU), average precision (AP over ellipse IOU threshold) [31], $AP^{\theta}$ (AP over angle error under ellipse IOU threshold), and F-1 Score. The expression of the F-1 Score is as follows:

$$\text{F}-1\text{Score} = \frac{2 \times \text{precision} \times \text{recall}}{\text{precision} + \text{recall}} \quad (8)$$

In addition, in industrial applications, apart from the aforementioned accuracy metrics, we are also concerned with whether all the ellipses in the test images are correctly detected. Therefore, we introduce the following metrics to further evaluate the performance of our model [32]:

$$\text{Reliability} = \frac{\substack{\text{Total number of test images with ellipses} \\ \text{presented all been correctly detected}}}{\text{Total number of test images}} \quad (9)$$

During the dataset testing, we obtain the AP values of our model by varying the ellipse IOU threshold from 70 to 90 with an interval of 5. We obtain a total of five AP values and computed their average, denoted as $AP_*$. Similarly, our $AP^{\theta}$ from the angle error of $45°$ to $5°$, with an interval of $5°$, resulting in 9 $AP^{\theta}$ values. We take their average and denote it as $AP_*^{\theta}$.

*4.3. Performance*

4.3.1. Module Validation and Comparative Experiments

In this section, we consider the adopted proposal extension operation and the improved loss function as two fundamental modules for conducting ablation experiments. These experiments aim to individually assess their impact on the detection accuracy of our model. The results of specific ablation experiments can be seen in Table 1.

**Table 1.** The performance of ablation experiments on different modules is evaluated. E: Proposal extension module. L: Improved loss function module. The default ellipse IOU for $AP^{\theta}$, F-1 Score, and Reliability is 0.90.

| E | L | MeanIOU | $AP_*$ | $AP_{80}$ | $AP_{90}$ | $AP_*^{\theta}$ | $AP_{30}^{\theta}$ | $AP_{20}^{\theta}$ | F-1 | Reliability |
|---|---|---------|--------|-----------|-----------|------------------|---------------------|---------------------|-----|-------------|
| - | - | 88.12 | 77.49 | 90.38 | 36.24 | 27.10 | 29.72 | 29.45 | 51.04 | 50.76 |
| ✓ | - | 88.71 | 80.28 | 90.18 | 49.25 | 33.19 | 39.33 | 36.21 | 60.24 | 60.12 |
| - | ✓ | 88.76 | 80.39 | 90.37 | 49.82 | 34.40 | 40.36 | 37.55 | 59.80 | 58.31 |
| ✓ | ✓ | 89.54 | 80.93 | 90.51 | 51.76 | 36.79 | 46.45 | 37.19 | 64.59 | 64.05 |

From Table 1, it is evident that the joint utilization of both modules results in a MeanIOU of 89.54% and an $AP_*$ of 80.93%. This showcases a 1% improvement in MeanIOU and a notable 3% enhancement in $AP_*$, compared to the scenario where neither module is employed. Furthermore, compared to the isolated application of each module, there are also noticeable enhancements. Moreover, considering the F-1 Score and Reliability metrics, their values rise to 64.59% and 64.05%, respectively, when both modules are employed. This signifies a substantial 13% and 14% boost, respectively, in comparison to their absence. In contrast to using only one module, a noteworthy 4% to 6% elevation can be observed in both metrics. The significant improvement in F-1 Score and Reliability

metrics demonstrates the effectiveness of our method in enhancing the robustness of model predictions and accurately detecting all objects in industrial images.

Regarding the specific performance on $AP^\theta$, it can be seen in Figure 8 that our method surpasses others when both modules are in use. This suggests that the enhancements introduced by our proposed method indeed have a positive impact on the accurate prediction of the ellipse rotation angle. Furthermore, the experimental results indicate that each improvement is indispensable and collectively contributes to the overall enhancement of performance.

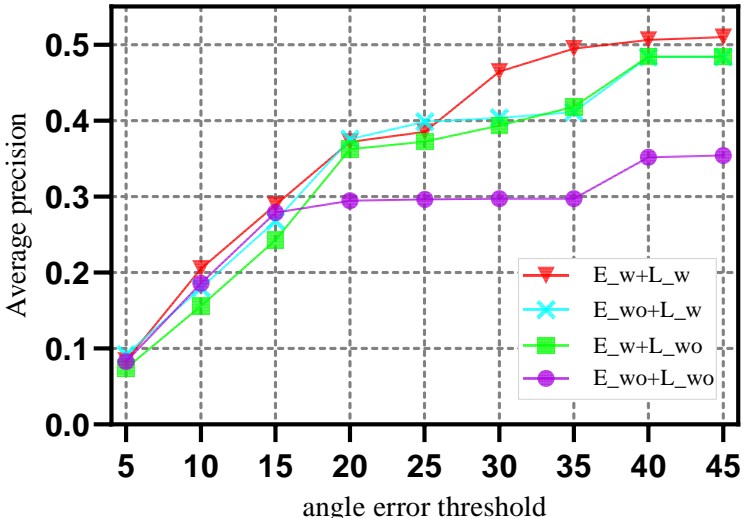

**Figure 8.** Comparison of $AP^\theta$ for different modules at various angle error thresholds. E_w: Proposal extension module is included. L_w: Improved loss function module is included.

During the comparative experiments on the test sets, we first compare our model with the classic object detection network, Mask R-CNN, in terms of several metrics. We discard the mask prediction branch of Mask R-CNN and replace the box regression with the regression of five ellipse parameters. The regression loss function is still the smooth $L_1$ function, keeping the rest of the structure unchanged. This network is considered our baseline model. In the process of the experiments, we compare our method not only with the baseline but also with the variations of the baseline network, such as replacing the regression loss function with the Gaussian angle distance or KLD. Furthermore, we also conduct comparative experiments using the hinge pins dataset on Ellipse R-CNN. Table 2 shows the specific results of the comparative metrics.

**Table 2.** The performance of our model compared with other methods is evaluated. The default ellipse IOU for $AP^\theta$, F-1 Score, and Reliability is 0.90.

| Methods | MeanIOU | $AP_*$ | $AP_{80}$ | $AP_{90}$ | $AP_*^\theta$ | $AP_{30}^\theta$ | $AP_{20}^\theta$ | F-1 | Reliability |
|---|---|---|---|---|---|---|---|---|---|
| Mask R-CNN (baseline) | 88.12 | 77.49 | 90.38 | 36.24 | 27.10 | 29.72 | 29.45 | 51.04 | 50.76 |
| Mask R-CNN (Gau) | 88.34 | 76.44 | 81.43 | 49.28 | 32.17 | 39.32 | 33.73 | 51.38 | 48.08 |
| Mask R-CNN (KLD) | 85.45 | 73.25 | 80.64 | 45.33 | 29.54 | 37.73 | 30.28 | 50.95 | 46.29 |
| Ellipse R-CNN | 86.99 | 80.39 | 89.38 | 45.50 | 33.59 | 45.50 | 34.61 | 53.70 | 51.92 |
| Our method | 89.54 | 80.93 | 90.51 | 51.76 | 36.79 | 46.45 | 37.19 | 64.59 | 64.05 |

From Table 2, our method shows certain advantages over other models in all the evaluation metrics. Our method achieves a MeanIOU of 89.54% and an $AP_*$ of 80.93%, which are improvements of 1% and 3%, respectively, compared to the baseline. Furthermore, our method achieves an $AP_*^\theta$ of 36.79%, which is a growth of 3~9% compared to the remaining four methods. This indicates that our method has an advantage in accurately regressing the ellipse rotation angle. As for the F-1 Score and Reliability metrics, our method

can reach 64.59% and 64.05%, respectively, showing a significant advancement of 13% and 14% compared to the baseline method. In the comparison experiments with other methods, the metrics we adopt demonstrate the effectiveness of our proposal extension operation and the combination of Gaussian angle distance with the smooth $L_1$ loss function operation.

Additionally, we also compute the average ellipse parameter estimation errors of our model and other models on the test sets. The statistical results can be seen in Table 3.

**Table 3.** The average ellipse parameter estimation errors.

| Methods | Position Error | Radii Error | Angle Error (°) |
|---|---|---|---|
| Mask R-CNN (baseline) | 3.02 | 2.37 | 26.18 |
| Mask R-CNN (Gau) | 3.05 | 2.30 | 26.27 |
| Ellipse R-CNN | 2.30 | 2.26 | 26.21 |
| Our method | 2.40 | 2.14 | 24.69 |

From Table 3, our method exhibits certain advantages over other methods in terms of radii and angle estimation errors.

### 4.3.2. Visualization Experiments

In this section, we conduct some visualization experiments. To validate that our method's detection performance is not affected by real industrial environments, we separately add the Gaussian noise and perform low-light processing on the images of the hinge pins in the test sets to simulate industrial environment noise. The specific detection results are shown in Figure 9. The experimental results demonstrate that our method can accurately detect the specific positions of the hinge pins even when the images are in a blurred or low-light state, indicating a certain level of robustness against interference.

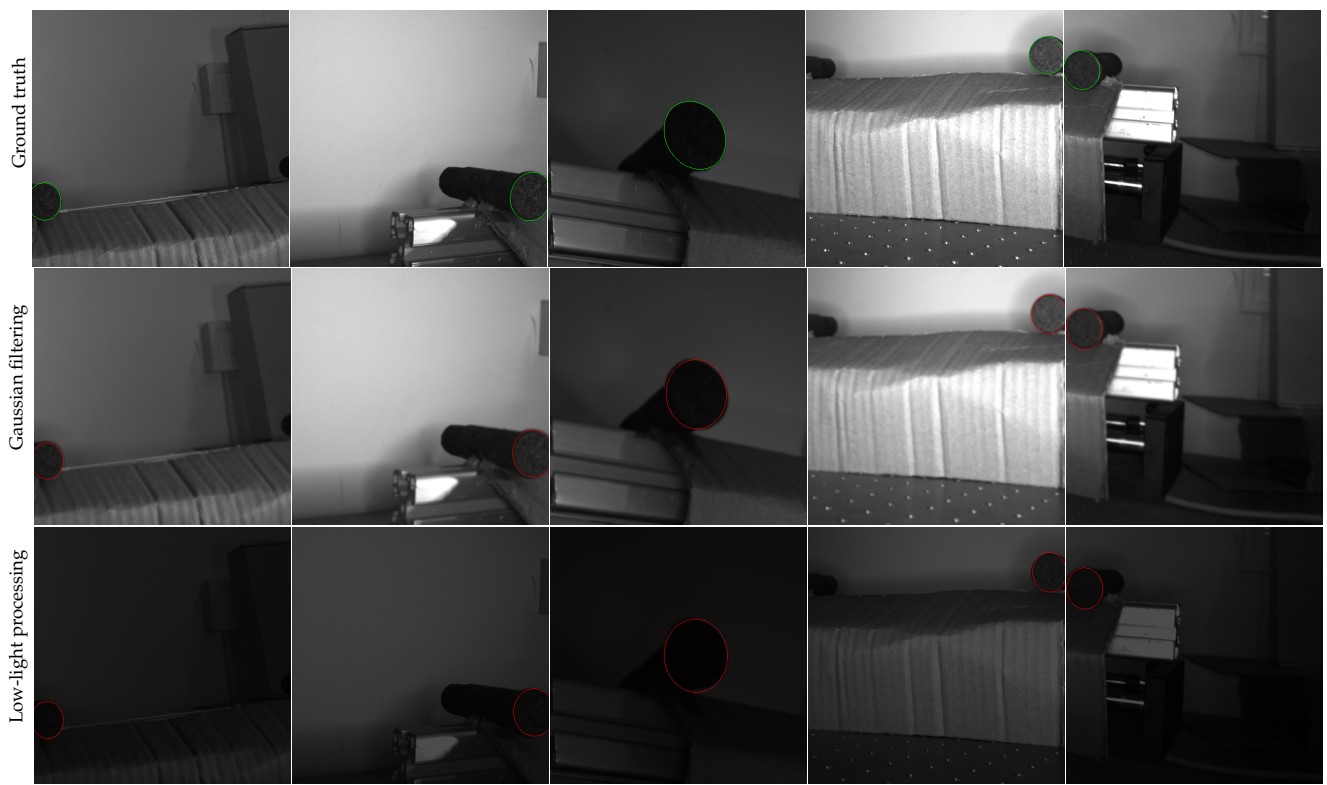

**Figure 9.** The detection performance of hinge pins under two simulated industrial environmental conditions. Green ellipses are the ground truth, and red are detected by our method.

We also conduct some experiments to provide a more detailed illustration of the performance of our method compared to Mask R-CNN and traditional ellipse detection methods on the hinge pins dataset, and the results are shown in Figure 10.

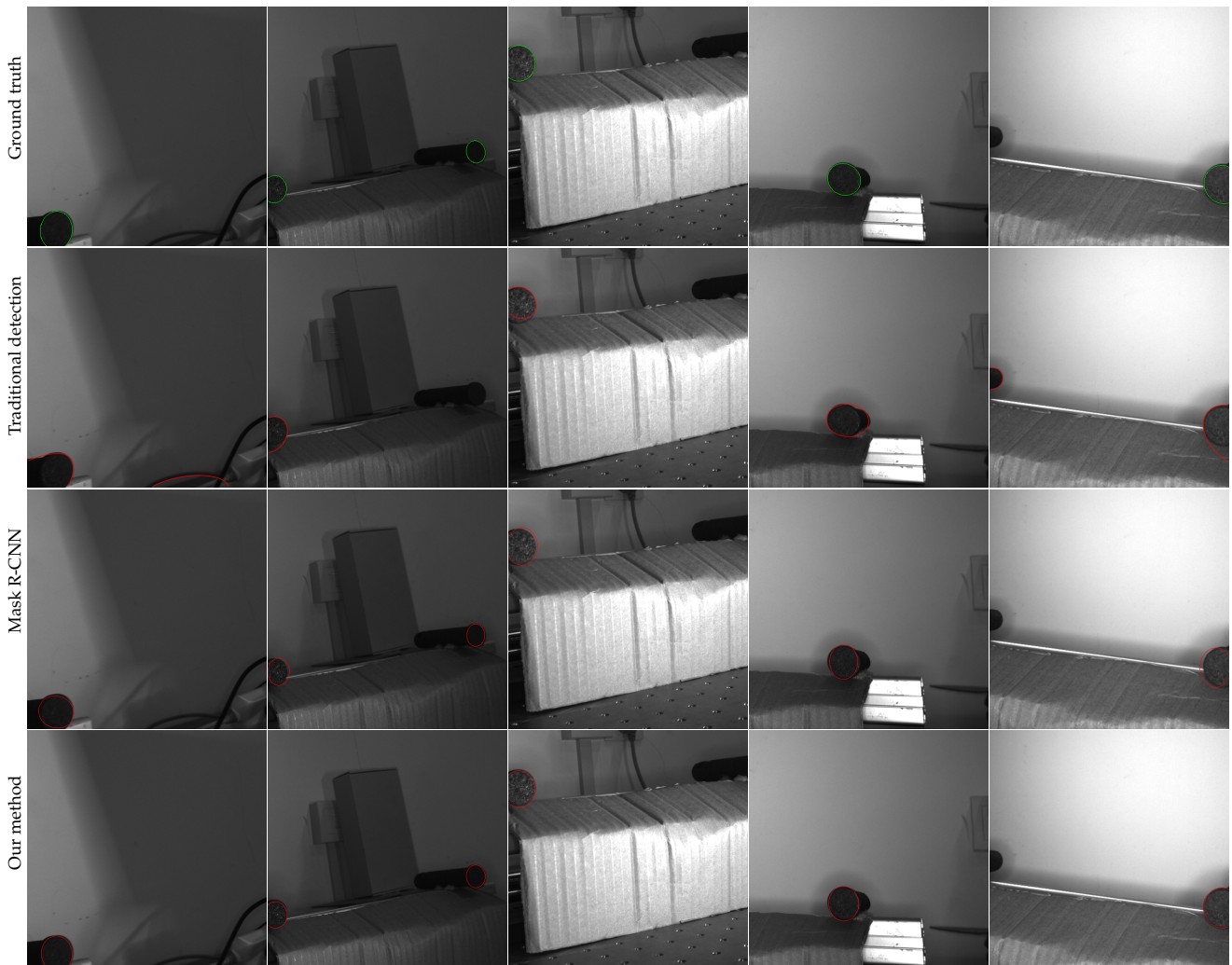

**Figure 10.** Examples of ellipses detected from the hinge pins dataset use Mask R-CNN (baseline), traditional detection method, and our method. Green ellipses are the ground truth, and red are detected by these methods.

In the traditional detection method, we adopt von Gioi's method [33] to extract subpixel edge contours of the ellipse in the image and perform ellipse fitting to obtain the hinge pins object in the image. By comparing it with the ground truth, the ellipse fitted by this method is greatly influenced by the presence of the background, resulting in numerous missed detections, false alarms, and inaccurate detections. Compared to our method, this approach is not very reliable in detecting hinge pins. Furthermore, when using Mask R-CNN for ellipse detection on the hinge pins dataset, the major issue is the inaccurate detection of incomplete ellipse occurring at the image boundary. This method often exhibits deviations and is unable to accurately detect such cases. Our method addresses the issue of inaccurate regression by extending the proposal. Additionally, we enhance the accurate prediction of ellipse rotation angle by incorporating Gaussian angle distance with smooth $L_1$ loss as the loss function for this regression task. These visualization results demonstrate that our method has better performance in accurately predicting all parameters of the ellipse.

## 5. Conclusions

In this paper, we propose a CNN-based method for ellipse detection in industrial images. An extension proposal operation is introduced to ensure accurate regression for an incomplete ellipse located at the image boundary. Additionally, by combining Gaussian angle distance and the smooth $L_1$ loss function, we further enhance the accurate prediction of the ellipse rotation angle. Due to the unavailability of a real hinge pins dataset and the constraints of the actual industrial scene, the simulation platform is set up in the laboratory to collect data using hinge pins. In subsequent research, on-site data will be further accumulated. A variety of experiments have been designed based on the existing data, including error estimation experiments and simulations under industrial environment conditions. These experiments demonstrate the effectiveness of our method for the automatic detection of hinge pin wear, which is of great significance for practical industrial production. Although our method has certain advantages, there is still scope for further improvement. In future research, we can acquire real industrial images of hinge pins in the metallurgical field to address practical industrial challenges. Additionally, our method can be further applied to other ellipse datasets to achieve a more comprehensive ellipse detection application.

**Author Contributions:** Conceptualization, K.L. and T.P.; methodology, K.L. and Y.T.; validation, K.L. and Y.T.; formal analysis, K.L. and Y.T.; investigation, K.L. and Y.T.; resources, Y.L., R.B. and K.X.; data curation, Y.L., R.B. and K.X.; writing—original draft preparation, K.L., Y.T. and Z.Z.; writing—review and editing, K.L., Y.T. and Z.Z.; visualization, K.L.; supervision, T.P. and Y.T.; project administration, K.L., Y.T. and Z.Z. All authors have read and agreed to the published version of the manuscript.

**Funding:** This research received no external funding.

**Data Availability Statement:** Not applicable.

**Conflicts of Interest:** The authors declare no conflict of interest.

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
