# Peer review of "Ellipse Detection with Applications of Convolutional Neural Network in Industrial Images"

_electronics, doi:10.3390/electronics12163431_

Round 1

Reviewer 1 Report

This paper proposes a CNN based method to detect elliptical shapes in industrial images. This version requires some improvements and rearrangements. Here are some suggestions:

 ·         The Introduction section would benefit from reformulation and rearrangements. For example, consider moving up the second paragraph on page 2.

·         Consider moving Table 1 and Figure 8 to after the first paragraph of Section 4.3.1.

·         Move Table 2 to after the first paragraph of Section 4.3.2.

·         Address how well the authors' method performs in different situations, such as low lighting or images with noise.

·         Consider combining Section 4.3.1 and Section 4.3.2 (the first two paragraphs) and add a new section to describe visual or qualitative analysis.

No remarks

Reviewer 2 Report

I have read manuscript entitled “Ellipse detection with applications of convolutional neural network in industrial images” in a fairly detailed fashion. This paper deals with convolutional neural network-based method for ellipse object detection. In general, the topic of this manuscript is academically and technologically relevant. However, there exist some points that need to be clarified.

Abstract should be modified in order to include only specific and interesting results for potential applications, and not give only a summary of the paper. The abstract must be short and concise!

The authors should mention explicitly what is novelty of this manuscript and what are the differences between it and the previous published studies which are available in the literature, as well as what is the goal of the author. In my opinion, a detailed performance comparison of the proposed solution with already available ones (with concrete citation of more references) in the form of a table is necessary. What are the advantages and disadvantages.

Considering the topicality of the subject, the references used in this manuscript are relatively old. There are a lot of papers from conferences instead of serious studies in high-ranking journals.

A careful revision of English is also recommended.

Based on the above, I think that manuscript in this version cannot be accepted for publication in Electronics since it needs a major revision.

A careful revision of English is recommended.

Reviewer 3 Report

This is a paper proposing a method to detect ellipses in "industrial" images. They make improvements on the Mask R-CNN to detect ellipses. They introduce a new loss function and show that it provides decent accuracy in some cases.

They compare their new method to some baseline R-CNNs and it out performs them. They mention "traditional" ellipse detection methods, but fall very short in that analysis. They focus on the edge extraction method and not the ellipse detection methods.

Their dataset does not seem like a useful dataset. It is very problem specific and doesn't look robustly assembled. The image in Figure 6 doesn't look related to the motivation image in Figure 1. 

They take an image and cut it into pieces. Some sub-images have pins and some do not. The image quality is terrible. Maybe it will be for the real industrial application as well, but their study configuration doesn't seem related to the industrial conditions.

They don't report average ellipse parameter estimation errors (error in position and radii), just saying detected or not detected and explore thresholds for that.

They don't describe how they come up with the "ground truth" estimates for the pin location and sizes. That needs to be in Section 4.1.

I don't think their results will be of interest to the wider scientific population because of the way they constructed their experiments and their dataset.

I don't know why they give a long discussion of the KLD method if they don't use it.

2048x2048 and 512x512 are dimensions, not resolutions.

The English is ok. The misplaced commas and spaces in the author list had me worried (space after comma and not before comma). The text in the body was ok, with several extra and missing articles, especially in the earlier pages. 

"And" is a conjunction. It goes BETWEEN two clauses, and thus never starts a sentence.

Round 2

Reviewer 1 Report

The authors have addressed all the concerns.

Alright, minor spell checking required

Reviewer 2 Report

The abstract is still not short and concise!!! I note once again that the abstract be modified in order to include only specific and interesting results for potential applications, and not give a summary of the paper or worse to contain the general facts as an introduction of the paper.

A careful revision of English is also recommended again.

Reviewer 3 Report

Adding Table 3 is good. Thanks. That is the biggest improvement in this version of the paper.

Related to that:

p5 How much will the errors in the table resulting in mismatches such as shown in Figure 4a affect any downstream operations like your motivational issue? What does that say about the angle of the pin itself? If the pins are in holes in the chains, their x-y positions can't vary much. BIG ISSUE

p3-109: I'm not sure what detecting faces has to do with detecting ellipses. If they include detecting faces, then what about detecting any other item?

p3-118 this section talks about loss functions more than distance metrics. Similar but different. Change name.

p9-269 & 273  & 275 "resolution of 2048 × 2048 pixels".   AGAIN: 2048 × 2048  are dimensions NOT resolutions. A resolution is number of pixels divided by a distance like 2048 pixels/meter

p9 Fig 7. How will the ability to NOT find a hinge pin in a picture of a cardboard box (Fig 7b) help the classifier network to work in a real case of hinge pins in pictures with linkage chains? I'm not convinced your dataset is useful. BIG ISSUE

p9 - you still do not describe how you got the ground truth ellipse positions in the 1862 images. The yellow text is not about GT generation. Did you as humans look at the images that contained hinge pins and manually trace ellipses? What did you do to tell the network what values it was aiming for? How did you get the ellipses in Figure 9 top row? BIG ISSUE

Re classical methods: You discuss Hough methods being computationally expensive. There are other classical methods beyond Hough. You don't need to try all classical methods, but your discussion needs to show knowledge of the topic. It currently does not.

Many grammatical errors remain. They do not detract from overall readability, but they need to detect and fix them. Some bigger issues (these are not ALL the problems):

p12-369 we -> We

p13-372 von's -> von Gioi's 

References: Fix capitalization-venue names, R-CNN. 

fix "In Proceedings of the Proceedings"

Round 3

Reviewer 3 Report

There are still major flaws with the data and experimental design.

"we divide the original images with dimensions of 2048 × 2048 into smaller patches 270 by dividing them into equal quarters in width and height. One original image can yield 16 271 smaller images with dimensions of 512 × 512, among which one to two images contain the 272 hinge pins object we want to detect."

If I was detecting the letter A on a blank page with only a letter A, and we divided the page so many tiles were blank, not finding the A in the blank tiles would not be a difficult task. If we were to try to detect that same letter A on a page full of text and other letters, then finding the letter A would be an accomplishment. Your dataset is not representative of finding a pin in a picture with metal linkage because not finding it in the area of neutral wall or cardboard box is not a challenge.